# Tuning the dynamic range of bacterial promoters regulated by ligand-inducible transcription factors

Ye Chen [1], Joanne M.L. Ho[1], David L. Shis[1], Chinmaya Gupta[2], James Long[1], Daniel S. Wagner[1], William Ott[2], Krešimir Josić[1,2,3] & Matthew R. Bennett[1,4]

One challenge for synthetic biologists is the predictable tuning of genetic circuit regulatory components to elicit desired outputs. Gene expression driven by ligand-inducible transcription factor systems must exhibit the correct ON and OFF characteristics: appropriate activation and leakiness in the presence and absence of inducer, respectively. However, the dynamic range of a promoter (i.e., absolute difference between ON and OFF states) is difficult to control. We report a method that tunes the dynamic range of ligand-inducible promoters to achieve desired ON and OFF characteristics. We build combinatorial sets of AraC-and LasR-regulated promoters containing −10 and −35 sites from synthetic and *Escherichia coli* promoters. Four sequence combinations with diverse dynamic ranges were chosen to build multi-input transcriptional logic gates regulated by two and three ligand-inducible transcription factors (LacI, TetR, AraC, XylS, RhlR, LasR, and LuxR). This work enables predictable control over the dynamic range of regulatory components.

[1] Department of Biosciences, Rice University, 6100 Main Street, Houston, TX 77005, USA. [2] Department of Mathematics, University of Houston, 4800 Calhoun Road, Houston, TX 77204, USA. [3] Department of Biology and Biochemistry, University of Houston, 4800 Calhoun Road, Houston, TX 77204, USA. [4] Department of Bioengineering, Rice University, 6100 Main Street, Houston, TX 77005, USA. Ye Chen and Joanne M.L. Ho contributed equally to this work. Correspondence and requests for materials should be addressed to K.J. (email: josic@math.uh.edu) or to M.R.B. (email: matthew.bennett@rice.edu)

Synthetic gene circuits are constructed by rewiring transcription factors and promoters to create novel regulatory topologies[1–5]. Promoters, which initiate transcription in response to transcription factors and associated ligands, are typically derived from endogenous components of the host or related species to ensure compatibility with the host transcription machinery. However, transcription rates of native promoters and their responses to inducers vary widely, as they are tuned to respond at rates appropriate to their natural setting[6–8]. This potential incongruity can cause problems when constructing gene circuits, as the fold-change induction of a natural promoter may not allow the synthetic circuit to behave as designed.

The first step in tuning a promoter is generally to change its overall output, i.e., to increase or decrease the amount of protein produced. This can be achieved in several ways. For instance, the copy number of the gene can be changed by placing it on different plasmids or by integrating it multiple times within the chromosome. Libraries of constitutive promoters have also been assembled and tested to achieve target expression levels[9]. In addition, although not technically part of the promoter, the 5′ untranslated region can be manipulated to alter translation rates and hence protein production[9–12].

Many attempts have been made to engineer transcriptional systems that are better suited for use in synthetic gene circuits[13–18]. Tuning these regulatory pathways involves building and testing circuits with precharacterized −35 and −10 sites[19] and Shine–Dalgarno sequences[20] of varying strengths until the desired properties (e.g., leakiness, fold induction) are empirically achieved; however, this approach can be extremely labor-intensive and costly.

Furthermore, although these methods can help to control gene expression, they universally alter protein output regardless of inducer concentration and the dynamic range of gene expression may change in unpredictable ways. For example, if one were to increase the copy number of a gene to increase protein production, the rates of production increase for both the OFF state (in the absence of inducer) and the ON state (in the presence of inducer).

To date, hybrid promoters that respond to two or more transcription factors have been constructed in several ways, including: by encoding operator sites for multiple transcription factors into a single promoter[21,22]; by using two consecutive promoters, each with its own regulatory features and transcription start site[18]; or by engineering different transcription factors that bind to the same operator[23,24]. Others have engineered promoters that have altered overall production of either mRNA[25–27] or protein[10]. However, the dynamic range of these synthetic promoters can be small—i.e., their outputs often show only a small difference between OFF and ON states in response to signal ligands that bind the transcription factors. Despite these advances in promoter engineering, the dynamic range of synthetic promoters has been difficult to tune. This problem prohibits the facile construction of multi-layer synthetic gene circuits, which require the output dynamic range of an upstream regulator to be compatible with the input dynamic range of a downstream target. Of note, ligand-inducible transcription factor systems can be manipulated at the level of the transcription factor as well as at the level of the promoter. In prior work, the leakiness and inducibility of the ligand-inducible LacI repressor mutants have been quantified and classified[28]. Such manipulation at the level of the transcription factor is informative and compatible with our promoter engineering approach, and both can be performed in conjunction to enable facile tuning of the dynamic range of synthetic promoters. Additionally, experimental studies have shown that inducibility and dynamic range can be tuned by changing the copy number of the plasmid encoding the transcription factor[29,30], mutating the

transcription factors[31], changing the operator sequence[6], mutating the RNA polymerase[25], and reducing promoter crosstalk[32]. In addition, methods have been developed to calculate the probability of RNA polymerase binding at a promoter as a function of the number of regulatory proteins in the cell[33,34]. Recently, the insulation of minimal promoters was demonstrated to enable precise engineering and biophysical modeling of complex synthetic transcription circuits[35]. These approaches have inspired our study and are compatible with our promoter engineering approach to accomplish facile tuning of the dynamic range of promoters.

Here we built a library of *Escherichia coli* promoters that have a spectrum of dynamic ranges. To do so, we used a modular approach in which promoters were assembled from libraries of five main components: (1) the region upstream of the −35 site (in which an operator site for a transcriptional activator can reside); (2) the spacer region between the −10 and −35 sites (in which an operator site for a repressor can reside); (3) the −10 site; (4) the −35 site; and (5) the downstream region encoding the gene of interest. The fold-change induction of the promoters was tuned using a variety of −10 and −35 sites. We also developed a thermodynamic model that predicted the contribution of free energy of binding to the overall transcriptional initiation rate, which we measured in a fluorescence-based plate reader experiment.

Our prediction and identification of the dynamic range allowed us to identify the ideal promoters for building multi-input hybrid promoters that can be used to make transcriptional logic gates. To build transcriptional AND gates with high signal-to-noise ratios, we perused our screened promoter library to identify two members that exhibited high fold-change induction and used their −35 and −10 cores to build hybrid promoters that respond to a variety of small-molecule inducers. Each hybrid promoter exhibited robust AND gate behavior—i.e., strong expression in the presence of all inducers and negligible expression in the absence of one or more inducers.

Overall, our results provide a method for efficiently altering the dynamics range of ligand-inducible promoters. This ability is key for constructing synthetic gene regulatory circuits that require precise input and output relationships and will allow researchers to tune complex synthetic gene regulatory circuits in a facile fashion. This paper provides a simple, cost-effective means of engineering promoters that provide user-defined dynamic ranges, which will enable the fine-tuning of the metabolic flux within synthetic biological and chemical circuits inside living cells.

## Results

**Predictable promoter behavior by motif-based construction.** In growing *E. coli* cells, most promoters are regulated by $\sigma^{70}$, a housekeeping transcription factor that binds the −10 and −35 sites of a promoter and enables RNA polymerase to bind and initiate transcription[36]. Transcription rates have been shown to be highly dependent on the sequences of the −10 and −35 sites for promoters regulated by $\sigma^E$ and $\sigma^{70}$, and certain −10 and −35 site combinations are associated with known transcription rates[19,37,38].

Here we characterize the relationship between the free energy of $\sigma^{70}$ binding to the −10 and −35 sites and the dynamic range of a library of promoters. We first formulated a thermodynamic model for transcriptional initiation based on the probability of $\sigma^{70}$ binding to DNA[13–18,34,39–41]. First, predicted transcription rates were derived from the probability of $\sigma^{70}$ binding, with the assumption that, in the presence of inducer, the probability of activator and $\sigma^{70}$ both binding DNA is much higher than the probability of $\sigma^{70}$ alone binding. Second, to describe the fold change in expression upon induction, the logarithms of the

predicted transcription rates were fitted to the experimentally measured transcription rates, with the assumption that recorded fluorescence is proportional to the transcription rate. To account for the different regulatory architectures in this work, this thermodynamic model was modified to consider the additional binding states of the different transcription factors. Specifically, thermodynamic models were made for single (activator alone) and multiple (activator and repressor) transcription factors with different states of binding (+/+, +/−, −/+, −/−) modeled in order to capture the entire range of possible biochemical interactions (see Supplementary Note 1 for more details).

Our model predicted that the dynamic range of a promoter could be tuned by varying the sequences of the −10 and −35 sites of the promoter (Fig. 1). We assumed that changes at these sites affected the equilibrium constant, $K_{eq}$, of $\sigma^{70}$ binding to DNA: If $K_{eq}$ is low, $\sigma^{70}$ binds poorly and the promoter will exhibit low leakiness but poor induction (bottom curve in Fig. 1b). If $K_{eq}$ is high, $\sigma^{70}$ binds tightly and the promoter will exhibit good induction but high leakiness (top curve in Fig. 1b). We postulated that, for a range of moderate $K_{eq}$ values, a promoter can be made to exhibit low leakiness and high induction, i.e., a large dynamic range (middle curve in Fig. 1b).

To test our hypothesis that the dynamic range of a ligand-inducible promoter could be tuned by varying the $K_{eq}$ values of the −35 and −10 sites, we constructed a library of promoters (Fig. 2a), each of which contained: (1) a proximal operator-binding site for either AraC or LasR (immediately upstream of the −35 site); (2) various −10 and −35 sites derived from E. coli promoter consensus sequences, native promoters, synthetic promoters[13,14], and arbitrary sequences; and (3) a fixed downstream reporter sequence encompassing the +1 site, a ribosome-binding site and the gene encoding yellow fluorescent protein (YFP). AraC and LasR were chosen because they are both ligand-inducible transcriptional activators but come from two different transcription factor families[42,43]. In addition, a LacI-binding site (LacO$_1$) was encoded in the spacer region between the −10 and −35 sites to examine the influence of repressors on the behavior of these promoters.

Each promoter in the library was cloned into identical plasmid backbones (pMB1 origin, kanamycin resistance) and transformed into E. coli strain CY015, which is a ΔlacI ΔaraC strain that also constitutively expressed genomically encoded araC and lasR. We expected the promoters to be activated upon addition of their corresponding inducers, i.e., arabinose for AraC, and 3O-C12-HSL for LasR. To measure the relative promoter units (RPU), we used a standardized method that has been adopted by the research community[19,44]. We measured the transcription rate of each promoter in the library with and without inducer and observed that rankings of the −10 and the −35 sites were consistent between AraC- and LasR-regulated promoters regardless of the identity and presence of each inducer (Fig. 2b). For certain combinations of −10 and −35 sites, the promoters achieved a maximum expression level (~10$^5$ RPU)—an observation that is consistent with that of prior reports[14,44]. Fold-change

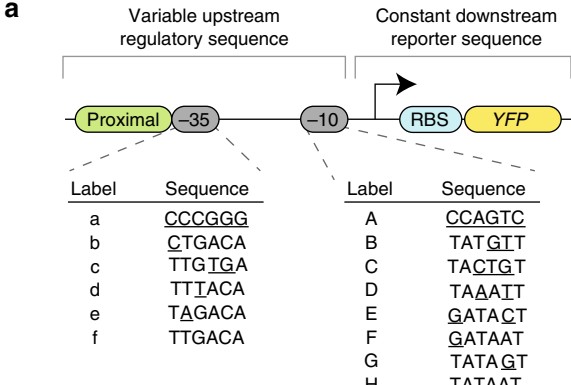

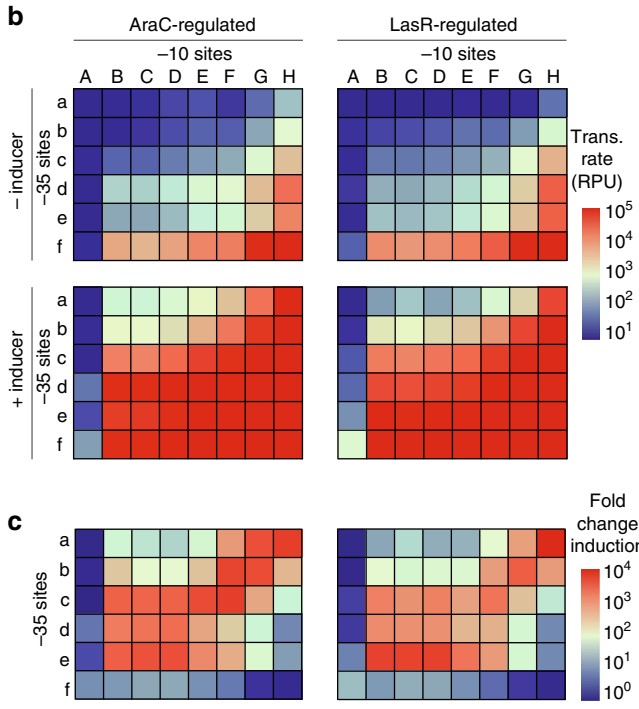

Fig. 2 Libraries of −10 and −35 sites cover a wide range of expression. **a** Diagram of the promoters showing essential features. Each promoter contained a proximal operator site for either AraC or LasR, a constant downstream reporter (YFP), and −10 and −35 sites chosen from the lists below. In each list, bases that do not match the consensus sequence are underlined. **b** Experimentally measured transcription rates of promoters regulated by either AraC (left column) or LasR (right column) with (bottom row) and without (top row) inducer for every combination of the −10 and −35 sites listed in **a**. **c** Fold-change induction heat maps for either AraC (left column) or LasR (right column). Experiments were performed with biological triplicates. Experimental data are available in Supplementary Data 1-3

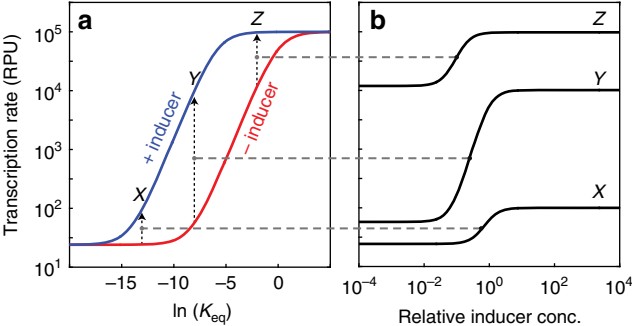

Fig. 1 Ligand-inducible promoters have different dynamic ranges. **a** Theoretically predicted transcription rate of a ligand-inducible promoter as a function of the relative equilibrium constant of σ-factor binding to the −10 and −35 sites, $\ln(K_{eq}) = -(\Delta G_{-10} + \Delta G_{-35})$ (see Supplementary Information). **b** Theoretically predicted transcription rates as a function of inducer concentration for three different but fixed values of the combined free energy of the −10 and −35 sites (as shown in **a**). The experimentally measured dynamic ranges of ligand-inducible promoters are shown in Supplementary Figure 1

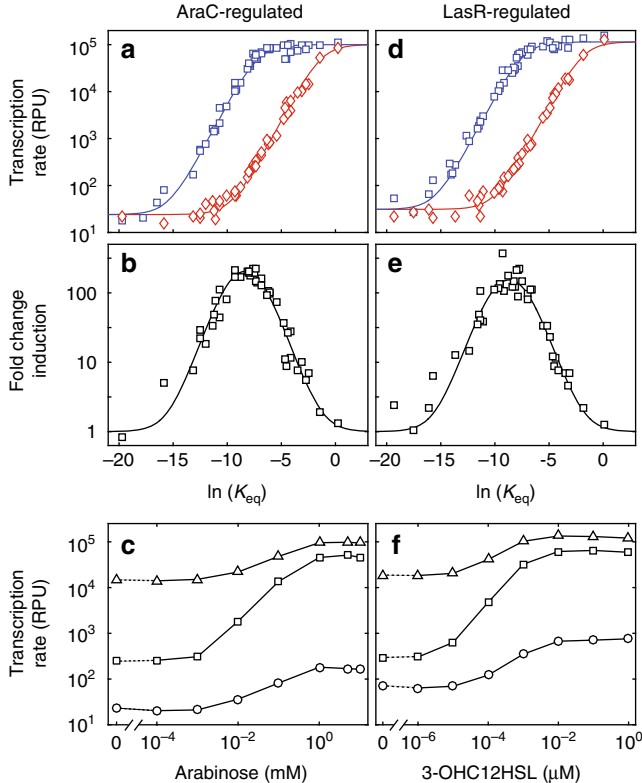

**Fig. 3** A thermodynamic model of promoter affinity fits the data. **a** Experimentally measured transcription rates of the AraC-regulated promoter as a function of the relative equilibrium constant of σ-factor binding to the −10 and −35 sites for both induced (blue squares) and uninduced (red diamonds) conditions. The thermodynamic model (solid curves) was fit to the results of each combination of the −10 and −35 sites. **b** The fold-change induction (ratio of the induced and uninduced transcription rates) of the AraC-regulated promoters as a function of relative equilibrium constant of σ-factor binding to the −10 and −35 sites. **c** Transcription rate as a function of arabinose concentration for three AraC-regulated promoters: $P_{ara-bD}$ (circles), $P_{ara-dE}$ (squares), and $P_{ara-eG}$ (triangles). **d** Experimentally measured transcription rates of the LasR-regulated promoter as a function of the relative equilibrium constant of σ-factor binding to the −10 and −35 sites for both induced (blue squares) and uninduced (red diamonds) conditions. The thermodynamic model (solid curves) was fit to the results of each combination of the −10 and −35 sites. **e** The fold-change induction (ratio of the induced and uninduced transcription rates) of the LasR-regulated promoters as a function of relative equilibrium constant of σ-factor binding to the −10 and −35 sites. **f** Transcription rate as a function of 3OH-C12-HSL concentration for three LasR-regulated promoters: $P_{las-bD}$ (circles), $P_{las-dE}$ (squares), and $P_{las-eG}$ (triangles). Experiments were performed with biological triplicates. Error bars in each plot have been omitted for clarity as they are, in most cases, smaller than the size of the symbols. Thermodynamic modeling data and experimental data are available in Supplementary Information and Supplementary Data 4, respectively

induction heat maps were consistent between AraC- and LasR-regulated promoters (Fig. 2c), indicating that the dynamic ranges of ligand-inducible promoters are predominantly controlled by the sequences of the −35 and −10 sites.

To better understand the behavior of the promoters within the library, we fit our thermodynamic model to the experimentally observed transcription rates (see Supplementary Information). The experimental results supported the predicted effects of $K_{eq}$ on transcription rates (Fig. 3a, d) and fold-change induction (Fig. 3b, e). Of note, promoters exhibit a range of leakiness and

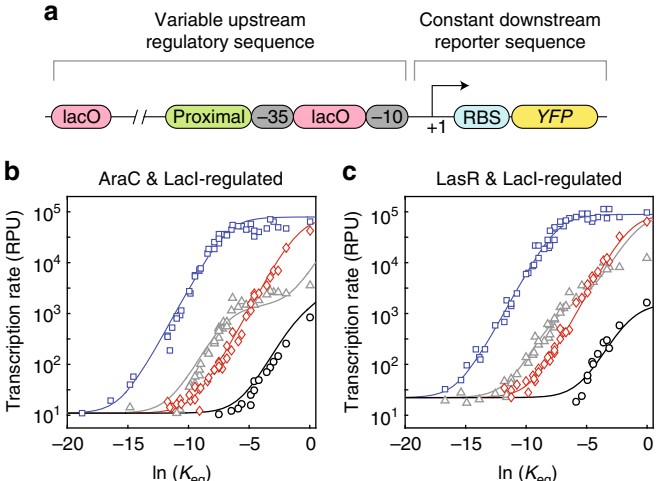

**Fig. 4** Hybrid promoters are tunable. **a** Diagram of the hybrid promoters showing essential features. Each promoter contained a proximal operator site for either AraC or LasR, the −10 and −35 sites from the library shown in Fig. 2, the constant reporter sequence, and two operator sites for the transcriptional repressor LacI. **b** Transcription rate of the AraC- and LacI-regulated hybrid promoters as a function of the relative equilibrium constant of σ-factor binding to the −10 and −35 sites for four different conditions: neither inducer (black), only arabinose (gray), only IPTG (red), and both arabinose and IPTG (blue). The thermodynamic model (solid curves) was fit to the results of each combination of the −10 and −35 sites. **c** Same as **b**, but for the LasR- and LacI-regulated hybrid promoters and with 3OH-C12-HSL instead of arabinose. Experiments were performed with biological triplicates. Error bars have been omitted for clarity as they are, in most cases, smaller than the size of the symbols. Thermodynamic modeling data and experimental data are available in Supplementary Information and Supplementary Data 5

inducibility depending on the free energy of σ[70] binding, as shown in the induction curves for three different combinations of −10 and −35 sites (bD, dE, and eG) in the AraC- (Fig. 3c) and LasR-regulated (Fig. 3f) promoters.

To demonstrate that our method applies not only to activators but also extends to other mechanisms of transcriptional regulation such as repression, we next investigated the behavior of hybrid promoters that respond to both an activator (either AraC or LasR) and a repressor (LacI) (Fig. 4a). To this end, we tested our promoter library in strain CY012, which is genetically identical to CY015 with the exception of a constitutively expressed genomically encoded *lacI*. The promoters were characterized under four conditions: in the presence of (a) neither inducer, (b) inducer of the activator only (arabinose or 3O-C12-HSL), (c) inducer of the repressor only (isopropyl β-d-1-thiogalactopyranoside (IPTG)), and (d) both inducers. The results were similar to those obtained in the absence of *lacI*—the ordering of −10 and −35 sites based on measured transcription rates was consistent under all four conditions. This data was also consistent with our two-input thermodynamic model in which extra states were included to account for repressor binding (Fig. 4b, c; Supplementary Information).

**Engineering responsive hybrid promoters**. To demonstrate the utility of our predictive model and experimental dataset for the construction of synthetic ligand-inducible promoters that behave as desired, we opted to perform a logic operation that required high inducibility and low leakiness (Supplementary Figure 3). Thus we decided to build AND gates using −35 and −10

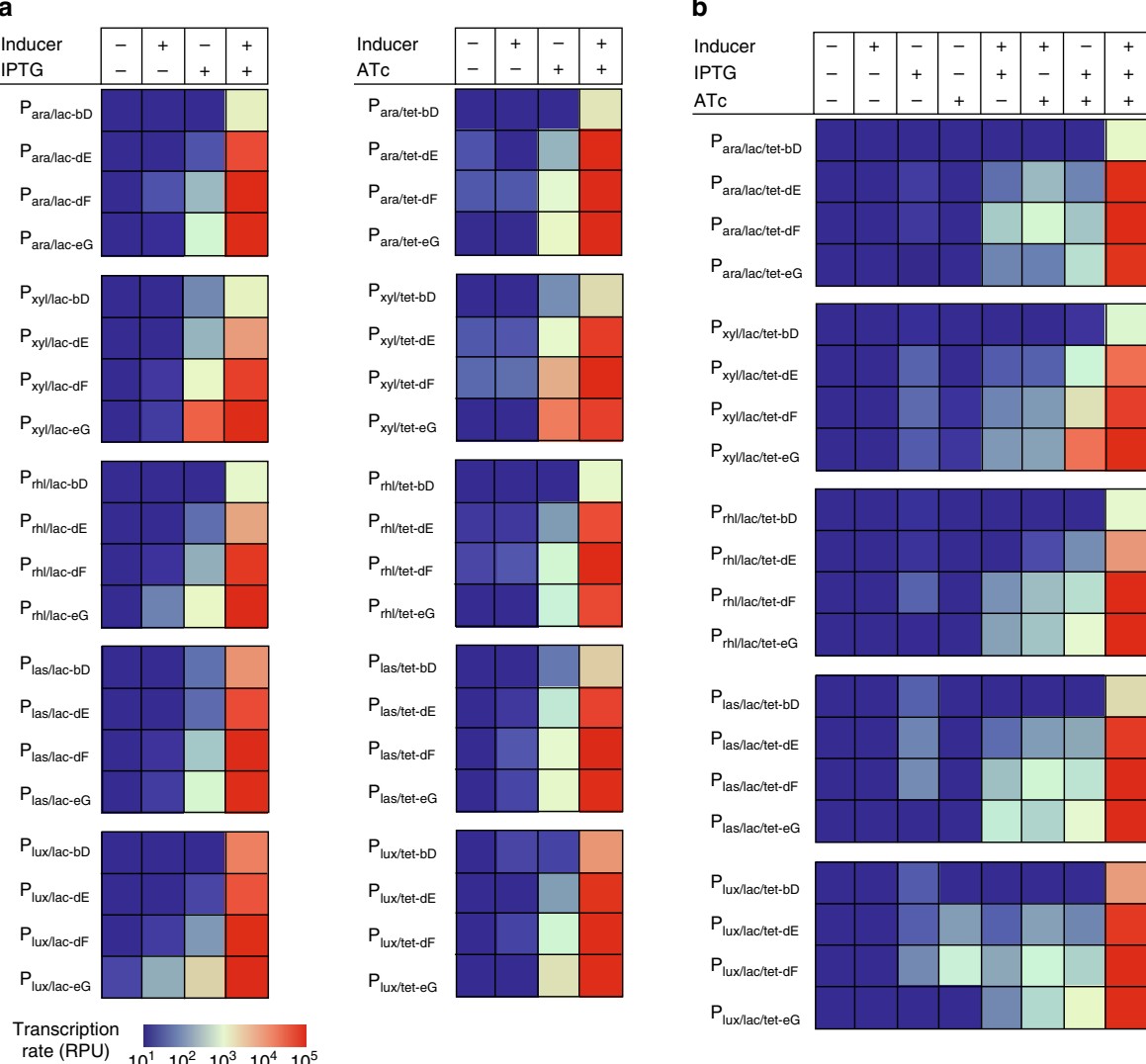

**Fig. 5** Combinations of −10 and −35 sites with large dynamic ranges are compatible with diverse regulatory sequences. **a** Experimentally measured transcription rates of a collection of engineered two-input hybrid promoters that use the dE, dF, bD, or eG combination of −10 and −35 sites. The promoters contain operator sites for one of the five different transcriptional activators (AraC, XylR, RhlR, LasR, or LuxR) and one of the two transcriptional repressors (LacI or TetR). Each promoter was tested in all four possible combinations of inducers for its respective activator and repressor. The AND gate is considered well behaved if the promoter is active only in the presence of both inducers. **b** Experimentally measured transcription rates of a collection of engineered three-input hybrid promoters that use the dE, dF, bD, or eG combination of −10 and −35 sites. These promoters contain operator sites for one of the five different activators (AraC, XylR, RhlR, LasR, or LuxR) and the operator sites for both LacI and TetR. Each promoter was tested in all eight possible combinations of inducers for its respective activator and repressor. Transcription should occur if and only if the inducers for all three transcription factors are present. Experiments were performed with biological triplicates. Experimental data for negative controls are available in the Supplementary Information (Supplementary Figure 2). Experimental data are available in Supplementary Data 6–8

combinations that were predicted and observed to exhibit the largest dynamic range. We assembled hybrid promoters that are tightly controlled by multiple ligand inputs beyond those tested above (Fig. 5). Specifically, these hybrid promoters contained operator sequences (upstream of the −35 site) that bind a variety of ligand-inducible transcriptional activators: AraC (arabinose), XylR (xylose), RhlR (C4-HSL), LasR (3O-C12-HSL), and LuxR (3O-C6-HSL). Additionally, we included operator sites for either LacI (LacO$_{sym}$) or TetR (TetO$_2$) in the spacer region to allow repression by these proteins[6,13]. For the −35 site, we chose to characterize version "d" (TTTACA) further because it provided medium affinity for $\sigma^{70}$. First, we slightly modified version "d" (TTTACA) to TTTACT because the terminal nucleotide, T, (a) is the second most common base pair at that position[45], (b) creates a one base pair overlap with the LacO$_{sym}$ operator site, (c) is

known to decrease the leakiness of the promoters in the presence of the repressor[6,13], and (d) the nature of this DNA substitution mutation does not alter the binding affinity of $\sigma^{70}$ for this −35 site[25]. As for the −10 sites, we chose versions "E" (GATACT) and "F" (GATAAT), because both of these sequences possess a 2 bp overlap with the TetO$_2$-binding site (allowing facile integration of the TetO$_2$ site) while providing different dynamic ranges. We posited that the difference in dynamic ranges would manifest as differences in the leakiness (basal transcription rates) and inducibility (induced transcription rates) of the AND gates built with these sequences. Specifically, version E that exhibited lower leakiness and lower inducibility was expected to provide tighter repression and lower induction, while version F that exhibited higher leakiness and higher inducibility was expected to provide leakier repression and higher induction. Two additional

$LacO_1$ and $TetO_2$ sites were also added (one downstream and one ~400 bp upstream of the +1 site) to achieve further repression[46].

As desired, all combinations of the hybrid promoters exhibited high induction only in the presence of both inducers and low signal in the presence of neither inducer (Fig. 5a). As expected, the "dE" combination of −10 and −35 sites provided tighter transcriptional regulation than the "dF" combination, which provided higher induction but was slightly more leaky than promoters with the "dE" combination (Fig. 5a).

To demonstrate that the high-inducibility and low-leakiness properties of these −35 and −10 combinations extend to multi-input hybrid promoters that are more complex, we constructed hybrid promoters that respond to three different inputs. For each three-input hybrid promoter, we included the "dE" or "dF" site, an activator-binding site (for AraC, XylR, RhlR, LasR, or LuxR), a $LacO_{sym}$-binding site in the spacer region, and two $TetO_2$ sites at the +1 position and upstream (~ −400 bp) region. Of note, the $TetO_2$ site at the +1 position provided more repression capacity than the upstream $TetO_2$ site. We observed that these hybrid promoters exhibited high gene expression if and only if all three inducers were present (Fig. 5b). These hybrid promoters demonstrate the utility of our experimental dataset for identifying the optimal −35 and −10 combinations for engineering gene expression.

Finally, to demonstrate the utility of our method for engineering synthetic promoters with desired dynamic ranges, we tested the bD and eG promoters, which possess distinct ON and OFF characteristics compared to dE and dF. Specifically, in contrast to the low-leak and high-signal performance of dE and dF (Supplementary Figure 3), promoter bD has low leak and low signal while promoter eG has high leak and high signal (Fig. 3c). As expected, the two-input (Fig. 5a) and three-input (Fig. 5b) hybrid promoters that contain the bD and eG sequences exhibit low leak and low signal as well as high leak and high signal, respectively (Fig. 5).

## Discussion

To date, synthetic gene circuits have been constructed primarily by cobbling together regulatory parts drawn from disparate pathways. For instance, the original genetic toggle switch[1] contained the repressors LacI (which regulates lactose metabolism) and TetR (which regulates the response to tetracycline). Because the lac and tet systems evolved separately, the promoters that respond to LacI and TetR are tuned to transcribe downstream genes at rates appropriate to their original setting. This tuning incongruity can cause problems for synthetic biologists, as the transcriptional response of a promoter needs to match the synthetic context of the circuit. Unfortunately, traditional strategies of testing −35 and −10 combinations and Shine–Dalgarno sequences of varying strengths to empirically achieve desired properties (with respect to, say, leakiness and inducibility) can be extremely time-consuming and labor-intensive.

In this study, we developed a method for tuning the difference between the ON and OFF states of a regulated promoter. We created a library of −10 and −35 sites that exhibit a wide range of fitted binding energies (inferred by fitting transcriptional data to a mathematical model) for $\sigma^{70}$ and characterized how the fitted binding energies affect the uninduced vs induced transcription rates of ligand-inducible promoters. Despite the relatively small number of examined −10 and −35 sites (a total of 48 combinations), our library members spanned the full range of possible input and output relationships, i.e., the data points span the entire breadth of the dynamic range curves shown in Figs. 3a, b and 4b, c. We further demonstrated the utility of our method for building synthetic gene circuits by using it to select optimal −35 and −10

combinations for building two- and three-input hybrid promoters with high inducibility and low leakiness.

Note that our model for transcriptional repression is not as quantitatively accurate as our model for transcriptional activation, possibly because our model does not account for additional biophysical states that may be assumed by the DNA during transcriptional repression (e.g., DNA looping[41]). Therefore, in the absence of inducers that bind the transcriptional repressors, the predicted transcription rates can deviate from the data for higher values of $\ln(K_{eq})$ (see gray plots in Fig. 4b, c).

Since the characteristics of each −10 and −35 site combination were independent of the transcription factor(s) being used to regulate the promoter (Figs. 3a, d and 4b, d), this approach should apply to the host of novel transcription factors that have been developed to regulate $\sigma^{70}$-based synthetic gene circuits[23,47] (provided the transcription factors have known ligands). Additionally, when used in conjunction with methods for controlling protein production rates, our approach should provide exquisite control over the dynamic range of gene expression in synthetic gene circuits.

## Methods

**Strains and plasmids**. We performed our experiments in *E. coli* strains derived from wild-type MG1655. To minimize interference of endogenous LacI with our exogenous LacI repressor, we performed lambda Red recombination[48,49] to prepare a LacI- strain—CY011 (*E. coli* strain MG1655 Δ*lacI*). In addition, since an *E. coli* LuxR homolog, *sdiA*, can partially activate $P_{rhl}$[50], we knocked out *sdiA* from CY011 to create CY013 (CY011 Δ*sdiA*). In order to test our −35 and −10 combinations in the context of three different promoters and transcription regulators, we built a plasmid pCH035 encoding wild-type *araC*, $P_{trc^*}$-*rhlR* (cloned from ATCC #47085), and $P_{lq}$-*lasR* (cloned from ATCC #47085) and performed lambda Red recombination to integrate these genes into the genome of CY013 to obtain CY015 (CY013 wt-*araC*, $P_{trc^*}$-*rhlR*, $P_{lq}$-*lasR*). Next, to test repressor function, we integrated $P_{lq}$-*LacI* and $P_{N25}$-*TetR* into the original LacI site of CY013 and CY015 via lambda Red recombination to obtain CY019 (CY013 $P_{lq}$-*LacI* $P_{N25}$-*TetR*) and CY021 (CY015 $P_{lq}$-*LacI* $P_{N25}$-*TetR*), respectively. After each recombination, antibiotic markers were removed by FLP-FRT recombination.

To construct the promoter library, we used a modified Golden Gate strategy[51] wherein BsaI was used to create a specific restriction site in the spacer region. The first segment contains the activator-binding site and the −35 site; the second segment contains the −10 site, a ribosome-binding site, and *yfp* (T203Y mutant of *sfGFP*) reporter gene[9]. After PCR amplification of each segment, all segments were ligated to the plasmid backbone ($Kan^R$ and pMB1 +*rop* origin) in all possible combinations. We used a similar strategy to construct the two- and three-input hybrid promoters. The genomic copy of XylR was used, but the majority of the activators were either integrated into the genome (AraC, LasR, and RhlR) or supplied on a separate plasmid (LuxR) ($Chlor^R$ and pSC101 origin).

All strains, plasmids, and promoter sequences used in this research are listed in Supplementary Tables 1–4. Key plasmids and strains are listed in Supplementary Table 5 and will be available by request on Addgene (https://www.addgene.org/Matthew_Bennett/) under the publication identifier corresponding to this manuscript.

**Promoter strength assay**. To measure the strengths of each −35 and −10 combination for promoters of the three activators, we first transformed the reporter plasmids encoding our library of promoters into strain CY015. We used the $P_{BAD}$, $P_{las}$, and $P_{rhl}$ promoters as reference standards against which we standardized the strengths of each member of the promoter library[52]. To measure the strengths of each −35 and −10 combination for promoters of the two repressors, we used strain CY021, a LacI, and TetR knock in strain. In the two- and three-input hybrid promoter assay, we transformed plasmids encoding the additional activator (LuxR) and hybrid promoters into CY015.

The strains were cultured overnight in LB media with 50 μg/mL kanamycin (and 25 μg/mL chloramphenicol as needed). The overnight cultures were inoculated at 1% volume per volume into 100 μL M9 media supplemented with 0.4% glycerol and 0.2% casamino acids and transferred into a 96-well plate. After growth with shaking at 37 °C for 2 h, 100 μL M9 media with 2× inducer was added to the cultures. The final concentrations of all inducers were as follows: C4-HSL (10 μM), 3O-C6-HSL (0.1 μM), 3O-C12-HSL (0.1 μM), arabinose (5 mM), xylose (5 mM), IPTG (1 mM), and ATc (100 ng/mL). Cultures were grown at 37 °C and 800 rpm in a plate shaker. The fluorescence of YFP was measured after 2 h using a plate reader (Tecan Infinite M1000; excitation 515 ± 5 nm, emission 528 ± 5 nm; PMT gain of 100) and reported as $Fluo_{test\_promoter}$ = (fluorescence$_{4h}$−fluorescence$_{2h}$) per $OD_{600}$[52]. Reference standards were included on each 96-well plate. The final

promoter strength was calculated in RPU using the equation $RPU_{test\_promoter} = Fluo_{test\_promoter} \times RPU_{reference\_standard}) \times Fluo_{reference\_standard}^{-1}$.

**Code availability.** Commented code and all collected coding data are available in the Github repository at https://github.com/josic/Promoter-Engineering.

**Data availability.** All experimental data are available in Supplementary Data 1–8 and are available from the authors upon reasonable request.

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

## Acknowledgements

We thank Dr. Mengyang Cao for his kind advice. This work was funded by the National Institutes of Health, through the joint NSF/NIGMS grant R01GM104974 (M.R.B., K.J., W.O.) and the NIGMS grant R01GM117138 (to M.R.B., K.J., W.O.); the National Science Foundation grant DMS-1122094 (to K.J.); and the Robert A. Welch Foundation grant C-1729 (to M.R.B.).

## Author contributions

Y.C. and M.R.B. conceived the experiment; Y.C., J.M.L.H., and D.L.S. conducted the experiments; J.L., D.S.W., W.O., and K.J. performed the thermodynamic modeling; all authors interpreted the data; Y.C., J.M.L.H., and M.R.B. prepared the manuscript.

## Additional information

**Competing interests:** The authors declare no competing financial interests.

