## [Peer Review File · Nature Communications]

Reviewers' comments:

Reviewer #1 (Remarks to the Author):

Overall: Below we note several typos as well as a few minor suggestions of the take-it-or-leave-it variety. However, we feel that the authors have addressed the majority of our concerns from their previous submission and believe the manuscript is now ready for publication.

Minor comments:

We suggest that you add a sentence emphasizing that the point of the paper is to provide a simple, low-cost means of engineering desired dynamic ranges for synthetic biology applications.

Please define dynamic range and fold-change/fold induction explicitly in the main text.

Since the central idea of this work is control over dynamic range, it would have been nice to see specifically how the dynamic range (and properties such as leakiness) vary with the various engineered constructs. On page 2, line 6, in the abstract, you note that you report the dynamic range of engineered promoters. I am unable to find any such values.

You could consider moving the extensive discussion of possible ways to tune the promoter (page 4, lines 3-16) to the Discussion section.

There are some supplemental figures that are never referenced, and hence it is difficult to discern their purpose. We suggest either referencing them in some form or removing them altogether.

In the rebuttal, multiple references are made to Brewster et al. as a validation of the fits/model. This analysis might be included in the SI.

Specific comments:

Pg 3, line 7: typo `... this approach can be extremely...`

Pg 4, line 10: typo `...changing the changing the...`

Pg 6, line 7: You could add more references about the relationship between transcription rate and probability of binding by RNAP (e.g. Ackers GK, Johnson AD, Shea MA. PNAS, 1982; Buchler NE, Gerland U, Hwa T. PNAS, 2003.; Vilar JM, Leibler S., J. Molecular Biology, 2003; Bintu L, Buchler NE, Garcia HG, Gerland U, Hwa T, Kondev J, Phillips R. Cur Op in Genetics & Development, 2005).

Pg 6, lines 12-16: We suggest you add the following sentence to line 12 as follows: `...to the transcription rate. To account for the different regulatory architectures in this work, this thermodynamic model was modified to account for the additional binding states of the different transcription factors. Specifically, thermodynamic models were made...`

Pg. 6, line 15: It is unclear what `reasoning` you are talking about.

Pg. 12, line 6-7: This is a very weak explanation, and in particular, no references are noted to support the hypothesis

Reviewer #3 (Remarks to the Author):

Thank you for your responses to my comments. Unfortunately, I still do not find the manuscript to be acceptable in its most recent form. As discussed below, some additional data is required to satisfy the concerns.

In response to point 6 regarding why the dE and dF site combinations were chosen, the authors state that, "Specifically, version E that exhibited lower leakiness and lower inducibility was expected to provide tighter repression and lower induction, while version F that exhibited higher leakiness and higher inducibility was expected to provide leakier repression and higher induction." This reasoning seems strange, as the heat maps in Figure 2b suggest that versions E and F have very similar performance.

The data regarding two- and three-input promoters in Figure 5 does little to support the rationale of the paper, i.e. that the generation of libraries of -10 and -35 sites enables the tuning of ligand-inducible promoters. It merely demonstrates that you can build AND gates with wide dynamic ranges. However, if a synthetic biologist needed an AND gate with specific ON/OFF behavior in order to match an input-output profile, and not necessarily a very wide dynamic range, how easily could they use this library to achieve the desired behavior? Testing the entire library of -10 and -35 sites in multi-input promoters is not necessary, but testing of a few select promoters with different ON/OFF characteristics is important.

Furthermore, the AND gates tested in Figure 5 lack meaningful comparisons. The dE and dF site pairs confer similar ON/OFF characteristics to single-input promoters, so it is not surprising that the dE- and dF-based multi-input promoters have similar induction characteristics. However, it is unclear how these dE- and dF-based AND gates perform against similar AND gates that had not been engineered in this fashion. We are unable to determine the improvement in dynamic range conferred by this style of library-based engineering.

The existing data are insufficient to address these concerns. Without the additional data discussed above, I cannot recommend this paper for publication in this journal.

Referee #1

1. *“In the attached review, we list a few minor suggestions of the take-it-or-leave-it variety. However, we feel that the authors have addressed the majority of our concerns from their previous submission and believe the manuscript is now ready for publication.”*

Response: Thank you. We have incorporated several of your suggestions. Of note, we now include this statement at the end of the introduction: “This paper provides a simple, cost-effective means of engineering promoters that provide user-defined dynamic ranges, which will enable the fine-tuning of the metabolic flux within synthetic biological and chemical circuits inside living cells. As for dynamic ranges of single-input and multi-input promoters, which are reported in Figs. 3 and 4, respectively, the supplementary tables also include the assayed values.

Reviewer #3

1. “Thank you for your responses to my comments. Unfortunately, I still do not find the manuscript to be acceptable in its most recent form. As discussed below, some additional data is required to satisfy the concerns.”

Response: We have performed the experiments that you requested and the additional data is included in Fig. 5 of our current re-submission.

2. “In response to point 6 regarding why the dE and dF site combinations were chosen, the authors state that, “Specifically, version E that exhibited lower leakiness and lower inducibility was expected to provide tighter repression and lower induction, while version F that exhibited higher leakiness and higher inducibility was expected to provide leakier repression and higher induction.” This reasoning seems strange, as the heat maps in Figure 2b suggest that versions E and F have very similar performance.”

Response: Regarding leakiness, the top-right heat map in Fig. 2b indicates that for the LasR-regulated promoters, dF is significantly leakier than dE. While this difference is not as stark in the top-left heat map for the AraC-regulated promoters, one can nevertheless observe from this heat map that dF is still leakier than dE. Regarding inducibility, the bottom-right heat map in Fig. 2b indicates that for the LasR-regulated promoters, dF is significantly more inducible than dE. In light of these observations, we respectfully disagree and stand by our original reasoning.

3. “The data regarding two- and three-input promoters in Figure 5 does little to support the rationale of the paper, i.e. that the generation of libraries of -10 and -35 sites enables the tuning of ligand-inducible promoters. It merely demonstrates that you can build AND gates with wide dynamic ranges. However, if a synthetic biologist needed an AND gate with specific ON/OFF behavior in order to match an input-output profile, and not necessarily a very wide dynamic range, how easily could they use this library to achieve the desired behavior? Testing the entire

library of -10 and -35 sites in multi-input promoters is not necessary, but testing of a few select promoters with different ON/OFF characteristics is important.”

Response: Thank you. Per your recommendation, we decided to further investigate bD (low leak/ low signal) and eG (high leak/ high signal). As you have requested, we have collected data for several promoters (bD and eG) with different ON/OFF characteristics for the five sets of inducible promoters (Ara, Xyl, Rhl, Las, and Lux) in the two-input (ATc or IPTG-inducible) and three-input (ATc and IPTG-inducible) configurations. The $2 \times 5 \times (2+1) = 30$ new data sets were collected for biological triplicates of each promoter configuration and the data is reported in Fig. 5. The assayed values are also provided in the Supplementary Information. We are grateful for your suggestion that we test additional promoters because our manuscript is now strengthened.

To summarize our findings, we have added a paragraph to our Results section stating: “Finally, to demonstrate the utility of our method for engineering synthetic promoters with desired dynamic ranges, we tested the bD and eG promoters, which possess distinct ON/OFF characteristics compared to dE and dF. Specifically, in contrast to the low leak/ high signal performance of dE and dF, promoter bD has low leak/ low signal while promoter eG has high leak/ high signal. As expected, the two-input and three-input hybrid promoters that contain the bD and eG sequences exhibit low leak/ low signal and high leak/ high signal, respectively (Fig. 5).”

4. “Furthermore, the AND gates tested in Figure 5 lack meaningful comparisons. The dE and dF site pairs confer similar ON/OFF characteristics to single-input promoters, so it is not surprising that the dE- and dF-based multi-input promoters have similar induction characteristics. However, it is unclear how these dE- and dF-based AND gates perform against similar AND gates that had not been engineered in this fashion. We are unable to determine the improvement in dynamic range conferred by this style of library-based engineering.”

Response: The focus of this paper is to provide a simple, low-cost method of engineering promoters with defined dynamic ranges for synthetic biology applications. In addition to providing this method, we also give an underlying theoretical explanation for how dynamic range of inducible multi-input promoters can be controlled. Our goal is not to engineer superior AND gates, but to help experimentalists make tunable AND gates with user-specified dynamic ranges. Experimentalists who use similar AND gates that had not been engineered in this fashion may measure the Relative Promoter Unit (RPU) of their promoter using Kelly *et al* (2009)*, wherein the authors reported a method for measuring Relative Promoter Unit (RPU) of any promoter by comparing its strength to a reference standard. By changing their -35 (or -10) sequence to one of those in our library, an experimentalist can easily identify the strength of their -10 (or -35) sequence by comparing RPU values. He or she can also make an informed decision as to which new -35 sequence to use in order to achieve a desired dynamic range.

*Kelly et al (2009) Measuring the activity of BioBrick promoters using an *in vivo* reference standard. *J. Biol. Eng.* 3: 4. DOI: 10.1186/1754-1611-3-4.

5. “The existing data are insufficient to address these concerns. Without the additional data discussed above, I cannot recommend this paper for publication in this journal.”

Response: As we have provided the additional data you requested, we trust that our revised manuscript is in good stead for publication in this journal.

REVIEWERS' COMMENTS:

Reviewer #1 (Remarks to the Author):

Thank you for performing the requested experiments. These data strengthen the manuscript considerably and have satisfied my concerns. I now consider this manuscript to be acceptable for publication.